# Exosome-delivered circRPS5 inhibits the progression of melanoma via regulating the miR-151a/NPTX1 axis

Haijun Zhu[ORCID][⊗]*, Pan Zhang[⊗], Jia Shi, Deqiang Kou, Xinping Bai

Department of Plastic Surgery, The Central Hospital of Wuhan, Tong ji Medical College, Huazhong University of Science and Technology, Wuhan, China

⊗ These authors contributed equally to this work.
* zhuhaijun9988@163.com

## Abstract

### Background

Circular RNAs (circRNAs) have been reported to exert critical functions in tumorigenesis and development. However, the underlying mechanism by which circRNAs regulate melanoma progression remain to be elucidated.

### Methods

The differentially expressed circRNAs were first identified by circRNA-seq, and circRNAs were validated via qRT-PCR and Sanger sequencing. Then, the impact of circRPS5, miR-151a and NPTX1 expression on the progression of melanoma cell were determined by gain- and loss-of-function assays. The relationship between circRPS5, miR-151a, and NPTX1 was predicted by StarBase website and authenticated by luciferase reporter assay. The melanoma cells-derived exosomes were characterized using nanoparticle tracking analysis (NTA) and western blot.

### Results

CircRPS5 was significantly downregulated in melanoma tissues and cell lines. Functionally, circRPS5 suppressed the proliferation, migration, and invasion of melanoma cells, and induced cell cycle arrest and apoptosis in vitro. Mechanistically, circRPS5 harbor miR-151a, acting as miRNA sponge, and then miR-151a targeted the 3'-UTR of NPTX1. Finally, circRPS5 was mainly incorporated into exosomes to inhibit the progression of melanoma cells.

### Conclusions

This finding reveal circRPS5 suppressed the progression of melanoma through miR-151a/NPTX1 pathway, and may provide a promising therapeutic strategies for melanoma.

**Data Availability Statement:** All relevant data are within the manuscript and its Supporting information files.

**Funding:** The author(s) received no specific funding for this work.

**Competing interests:** The authors have declared that no competing interests exist.

## Introduction

Melanoma is one of the most common malignant skin tumor originating from melanocytes with a highly aggressive and mortality [1]. Globally, the incidence of melanoma is rapidly increasing and is associated with ultraviolet (UV) irradiation, especially in fair-skinned populations receive excessive sun exposure [2]. The prognosis of melanoma remains dismal due to the insensitivity of melanoma cells to anticancer drugs [3–5]. Therefore, further research is urgently needed to explore the molecular pathogenesis of melanoma and identify more effective therapeutic targets for patients with melanoma.

Circular RNAs (circRNAs) are a type of endogenous noncoding RNAs (ncRNAs) with single-stranded closed-loop RNA molecules lacking terminal 5' caps and 3' poly-adenylated tails [6]. Owing to this extraordinary construction, circRNAs are resistant to exonuclease degradation and more stable than linear RNAs [7]. Moreover, circRNAs can function as competing endogenous RNAs (ceRNAs) and regulate gene expression at the transcriptional or post-transcriptional level by sponging microRNAs (miRNAs) [8, 9]. Thus, circRNAs play crucial roles in many cellular processes, and their dysregulation is involved in the occurrence and development of various disease, most prominently in cancer [10, 11]. For example, a novel circRNA named hsa_circ_0001666 was found to suppress the progression of colorectal cancer through the miR-576-5p/PCDH10 axis [12]. Huang et al. found that hsa_circRNA_104348 could act as a tumor promotor in hepatocellular carcinoma by sponging oncogenic miR-187-3p to promote RTKN2 expression [13]. However, the role and underlying mechanisms of circRNAs in melanoma remains to be unmasked.

Exosomes are a type of extracellular vesicles (30–150 nm in diameters) secreted from almost all living cells [14]. They contain endogenous proteins and nucleic acids that play essential roles in intercellular communications [15]. Mounting evidence suggests that exosomes play an essential role in tumor occurrence and progression through the crosstalk between cancer and non-cancer cells within the tumor microenvironment [16]. Recently, exosome-mediated transfer of circRNAs has been reported to play a crucial role in regulating tumor growth, metastasis, and angiogenesis in the process of cancer development [17, 18]. For instance, exosomal circRNAs derived from adipocytes promotes the tumorigenesis of hepatocellular carcinoma by regulating the miR-34a/USP7 pathway [19]. However, the relationship between exosomal circRNA and melanoma remains unclear.

In this study, we demonstrate that a novel circRNA named circRPS5 significantly downregulated in melanoma tissues and cell lines. Subsequently, we found that circRPS5 was negatively associated with melanoma progression by sponging miR-151a to influence the expression of neuronal pentraxin 1 (NPTX1). Moreover, we also found circRPS5 is mainly exist in exosomes and then plays a regulatory role in melanoma cells. Collectively, our findings suggest that circRPS5 might act as a promising therapeutic target for melanoma therapy.

## Methods

### Bioinformatics analysis

We first downloaded the raw transcriptome data (GSE31909, GSE35388 and TCGA-SKCM) and the corresponding clinical information from Gene Expression Omnibus (GEO) and TCGA database. the expression matrix is standardized according to the RMA and LIMMA algorithm [20]. Then, the differentially expressed genes (DEGs) between experiment and control group were identified using DESeq2 package with thresholds of logFC > 1.3 and P < 0.05 [21]. The candidate molecules were identified by overlapping DEGs with predicted targets and visualized using the VennDiagram package.

## Cell culture and transfection

The normal human melanocytes (HeMa-Lp) and human melanoma cells (A375 and A2058) were purchased from the Cell Bank of Type Culture Collection of Chinese Academy of Sciences China (Shanghai, China). All the cells were cultured in high-glucose Dulbecco's Modified Eagle Medium (DMEM, GibcoBRL, Invitrogen, Carlsbad, CA, USA) and supplemented with a 10% fetal bovine serum (FBS; Thermo Fisher Scientific, Waltham, MA, USA), 100 U/ml penicillin and 100 μg/ml streptomycin. During the culture period, the cells were incubated at 37°C in a humidified atmosphere of 5% $CO_2$ and 95% air.

The A375 and A2058 cells were seeded in 6-well plates at a density of $2 \times 10^4$ cells per well. The cells were then transfected with Lipofectamine 3000 (Invitrogen, Carlsbad, CA, USA). Briefly, pcDNA, miRNA inhibitor or mimic was diluted in Opti-MEM (Invitrogen, Carlsbad, CA, USA), stand for 5 min, and mixed with Lipofectamine 3000. Transfection solution was stand for 20 min and applied in each well. 72 h after the transfection, cells were harvested for the later study.

## Quantitative reverse transcription polymerase reaction (qRT-PCR)

Total cellular RNA was extracted from the cultured melanoma cells using TRIzol reagent (Invitrogen, Carlsbad, CA, USA), according to the manufacturer's instructions. Reverse transcription was performed using 1.0 μg total RNA and a MicroRNA Reverse Transcription Kit (Takara Biotechnology, Japan) or Prime Script™ RT kit (Takara, Dalian, China), which were used to investigate the expression of miRNA and mRNA, respectively. Amplification reactions was carried out on an ABI 7500 Sequencing Detection System (Applied Biosystems, CA, USA) with a SYBR green PCR Master Mix (Takara, Japan). All reactions were run in triplicate and were normalized to the miRNA house-keeping gene U6 or the mRNA house-keeping gene GAPDH. All primers are listed in Table 1.

## RNA fluorescent in situ hybridization (FISH)

The FISH assay, Cy3-labeled circRPS5 probes and 488-labeled miR-151a probes were designed and synthesized by RiboBio (Guangzhou, China). Briefly, the cells were fixed and then hybridized with the probe by incubation at 37°C overnight. After washing with phosphate-buffered saline (PBS) for three times, the cell nuclear were stained with DAPI. Finally, the images were taken using a Nikon A1 confocal microscope (Nikon, Japan).

## RNase R linear RNA digestion experiment

The total RNA was incubated with or without 40 U RNase R at 37°C for 15 min. After treatment with Rnase R, the expression levels of RPS5 and circRPS5 were analyzed using qRT-PCR.

## Cell counting kit-8 (CCK-8) assay

Cell viability in different groups was measured using CCK-8 assays (Dojindo, Japan). Briefly, $5 \times 10^3$ cells were seeded into 96-well plates, and then incubated with 10 μL of CCK-8 regent at indicated time point. After incubation, the absorbance value of each well was recorded at 450 nm using a microplate reader (Thermo, USA).

## Colony formation assay

The transfected melanoma cells in different groups were seeded into the 6-well plates at a density of $5 \times 10^2$ cells per well. After 2 weeks incubation, these plates were fixed by 10%

**Table 1. The sequence of PCR primers and oligonucleotide sets used for short hairpin RNAs, or probe.**

| Primer | Sequence |
|---|---|
| CircRPS5 | Forward: 5'- CTGAGTGCCTGGCAGATGA -3' |
| (convergent) | Reverse: 5'- AGGGCAGACAGGTTTATTGG-3 |
| CircRPS5 | Forward: 5'- GCCCAATAAACCTGTCTGCC -3' |
| (divergent) | Reverse: 5'- GCAATGGTCTTAATGTTCCGGA -3' |
| CircRPLP1 | Forward: 5'- GGGAGCCTCATCTGCAATG-3' |
| (convergent) | Reverse: 5'- CTCGGATTCTTCTTTCTTTGC-3' |
| CircRPLP1 | Forward: 5'- TGAGGAGAAGAAAGTGGAAGCA -3' |
| (divergent) | Reverse: 5'- AGGCTCCCAATGTTGACGTT -3' |
| miR-151a | Forward: 5'- GGATGCTAGACTGAAGCTCCT-3' |
| | Reverse: 5'- CAGTGCGTGTCGTGGAGT -3' |
| miR-324 | Forward: 5'- TGTGGTTACGGGATCCCCTACGC-3' |
| | Reverse: 5'- GCGTAGGGGATCCCGTAACCACA-3' |
| hsa_circ_0054602 | Forward: 5'- AGACTGATCTTTGCTGGCAA -3' |
| | Reverse: 5'- GGTGATGGTCTTCCCCGTAA -3' |
| hsa_circ_0020708 | Forward: 5'- TGAGAAGAAGGAGGAGTCTGA -3' |
| | Reverse: 5'- AGGTACACTGGCAAGCTTG -3' |
| hsa_circ_0064917 | Forward: 5'- TCGATGGACCTTGCACTCAA -3' |
| | Reverse: 5'- AAAGGAGACATAGGCCACCC-3' |
| hsa_circ_0007999 | Forward: 5'- TTTTGACCTGCTCCGTTTCC-3' |
| | Reverse: 5'- TGTTCAAAAGAGACGGGGTC-3' |
| hsa_circ_0076141 | Forward: 5'- GACATCGAGGCGCTGAAAA-3' |
| | Reverse: 5'- GGGAGTGGACTTAAGCCTGA-3' |
| hsa_circ_0011392 | Forward: 5'- TCAACTCAGCTGCCCTCTC-3' |
| | Reverse: 5'- GCGGTCACAAACATGGTCAT-3' |
| hsa_circ_0050747 | Forward: 5'- GACACGACACCCTGATCTGA-3' |
| | Reverse: 5'- ACCTCCATGTCATCTCCAGC-3' |
| hsa_circ_0078180 | Forward: 5'- ATCGCTGCAAATCTCCAACC-3' |
| | Reverse: 5'- AAGCCCCAATTACCTCCGTT-3' |
| NPTX1 | Forward: 5'- ACAGCCGCCTCAATTCCTC-3' |
| | Reverse: 5'- GCTCTTCTTCACCTTGGCATACA-3' |
| AURS2 | Forward: 5'- CACCTCCTCATCACAGCAACT-3' |
| | Reverse: 5'- TCTTCTCGACGCTTTCCCT-3' |
| RPS6KA2 | Forward: 5'- ATCCACTTCACCGATGGCTAC-3' |
| | Reverse: 5'- CGCATCAGCTCCATTACCAG-3' |
| KAT2B | Forward: 5'- TGTTCCTAAACCGCATCAA-3' |
| | Reverse: 5'- CGAGGTAGACTGTCGCAGA-3' |
| MYO5A | Forward: 5'- ATGTGGTCCGCAGGAGGTA-3' |
| | Reverse: 5'- AAGCAGCACTGAAGGTAGATG-3' |
| HMG20B | Forward: 5'- CGTGGTGACTGTCAAGCAAGAG-3' |
| | Reverse: 5'- AGATGGGAACATCGAAGGTGG-3' |

formaldehyde and stained with 0.5% crystal violet solution. The colonies were defined as > 50 cells/colony and photographed using an inverted microscope (Olympus IX71, Japan).

## 5-ethynyl-20-deoxyuridine (EdU) analysis

Cell proliferation ability was detected using the EdU assay kit (Ribobio, Guangzhou, China). Briefly, melanoma cells were seeded into 96-well plates at a density of $1 \times 10^4$ cells/well, and

then treated with culture medium containing 50 μM EdU reagent at 37°C for 2 h, fixed with 4% formaldehyde for 30 min. The nuclei were stained with Hoechst 33342. Finally, the results were photographed by a fluorescence microscope (Nikon, Tokyo, Japan), and the number of EdU-positive cells were quantified and analyzed.

## Transwell cell migration and invasion assay

The migration and invasion abilities of melanoma cells were assayed using Transwell inserts and Matrigel-coated Transwell (Corning, MA, USA). Briefly, $2 \times 10^4$ melanoma cells were suspended in 200 μL of medium free of serum, and added into the top chamber. 500 mL culture medium containing 10% FBS was added into the lower chamber. 24 h after incubation, a cotton swab was used to remove cells on the upper surface of the membrane gently and the cells were fixed in 4% paraformaldehyde, then 0.5% crystal violet was used to stain these cells for 20 min. The migrating or invading cells were photographed and counted under a microscope.

## Wound healing assay

For the migration assay, the transfected melanoma cells were seeded in 6-well plates at a high density and grew until ~80% confluence. The cell monolayer was scratched with a 200 μL sterile plastic tip, and then the cells were cultured in reduced serum culture medium at 37°C for 48 h. The movement of cells was photographed with a phase-contrast microscope.

## Western blotting

The total protein from cells was extracted using radioimmunoprecipitation assay (RIPA) lysis buffer (Beyotime, Shanghai, China) on ice for 30 min. The protein concentration was quantified by using a bicinchoninic acid (BCA) kit (Beyotime). Equal amounts of protein (20ug/well) were separated by SDS-PAGE gel and transferred onto a polyvinylidene fluoride (PVDF) membrane (Millipore, USA). After blocked in 5% non-fat milk for 1 h at room temperature, the membrane was then incubated with specific primary antibodies, including anti-NPTX1 (Proteintech, 20656-1-AP, 1:1000), anti-E-cadherin (Proteintech, 20874-1-AP, 1:1000), anti-N-cadherin (Abcam, ab76011, 1:1000), anti-TSG101 (Abcam, ab125011, 1:1000), anti-HSP70 (Abcam, ab2787, 1:1000), and anti-GAPDH (Proteintech, 60004-1-Ig, 1:5000) antibodies, followed by incubation with secondary horseradish peroxidase (HRP)-conjugated antibodies. After washes, signals were detected using ECL chemiluminescence reagent (Millipore, USA) and a chemiluminescence system (Bio-Rad, USA) and analyzed using Image Lab Software.

## Flow cytometry analysis

The melanoma cells in different groups were collected using trypsin (Thermo Fisher Scientific) and fixed for 8 h, Then, the cells were incubated with RNase (50 μg/ml) and propidium iodide (PI; 50 μg/ml) for 30 min. Finally, the cell cycle was determined by fow cytometry (BD Accuri C6) and analyzed using FlowJo software (TreeStar, OR, USA).

## RNA sequencing assay

Total RNA was extracted using TRIzol reagent (Invitrogen, MA, USA). Ribosomal RNA was removed from samples. The RNA-seq library was deep sequenced to identify the differentially expressed circRNAs at Novogene Bioinformatics Technology Co., Ltd.

## Terminal deoxynucleotidyl transferase (TdT)-mediated dUTP biotin nick end labeling (TUNEL) staining

Cell apoptosis was measured by using TUNEL assay kit (Beyotime, Nanjing, China). Briefly, the melanoma cells in 24-well plates were washed with PBS and fixed by 4% paraformaldehyde for 30 min at room temperature. Subsequently, the cells were incubated with TUNEL reaction mixture at 37˚C for 60 min in a dark, and the nuclear were stained with DAPI. Finally, TUNEL-positive cells were imaged under a light microscope (Olympus, Tokyo, Japan).

## Dual-luciferase reporter gene assay

The melanoma cells were seeded into 96-well plates and cultured to 50%–70% confluence before transfection. The wild-type (WT) or mutant type (MUT) circRPS5 and NPTX1 were inserted into pGL3-firefly luciferase vector, and then transfected into melanoma cells combination with miR-151a mimics or NC using Lipofectamine 3000 (Invitrogen, USA). After 48 h of transfection, the relative luciferase activity was conducted using Dual Luciferase Reporter Assay System (Promega, WI, USA).

## Exosome isolation and identification

Exosomes from cells were collected from culture media, and isolated by differential centrifugation and ultracentrifugation. Briefly, the media were first centrifuged at 800×g for 15 min at 4˚C, and then centrifuged at 10,000×g for 30 min at 4˚C to remove the redundant cells and the cellular debris. Finally, the exosomes were separated and pelleted by centrifugation at 100,000×g for 2 h at 4˚C. The exosome was resuspended in 100 μl of sterile 1× PBS. The nanoparticle tracking analysis (NTA) was used to detect the size and total amount of exosomes. The western blot assay was used to identify protein markers of exosome.

## Statistical analysis

All data are presented as the mean ± standard deviation (SD), and statistical analyses were conducted using GraphPad Prism 7.0 (GraphPad, USA) software. Non-paired t test was used to estimate the statistical differences between two groups. One-way analysis of variance (ANOVA) was used to determine the differences between three or more groups. All experiments were repeated at least three times, and $P < 0.05$ was statistically significant.

# Results

## CircRPS5 was aberrantly downregulated in melanoma tissues and cells

First, we applied next sequencing technology to identify 1342 differentially expressed circRNAs (DECs) between melanoma and para-tumor tissues (Fig 1A). Next, we verified the expression level of top 10 DECs in melanoma cells. The results showed that only 4 circRNAs (hsa_circ_0052351, hsa_circ_0036090, hsa_circ_0054602, and hsa_circ_0020708) showed four clearly visible strips, which suggested the possible existence of these four circRNAs in melanoma cells (Fig 1B). Schematic diagram illustrated the formation of hsa_circ_0036090 originated from RPLP1 pre-mRNA (exon 2, 3), and the formation of hsa_circ_0052351 originated from RPS5 pre-mRNA (exon 5, 6). Moreover, their head-to-tail splicing structure was confirmed by Sanger sequencing (Fig 1C). The circular properties of hsa_circ_0052351 (named circRPS5) and hsa_circ_0036090 (named circRPLP1) were identified with divergent primers and convergent primers, respectively (Fig 1D). qRT-PCR revealed that only circRPS5 expression was significantly downregulated in the two BC melanoma cell lines (Fig 1E). In A375 and

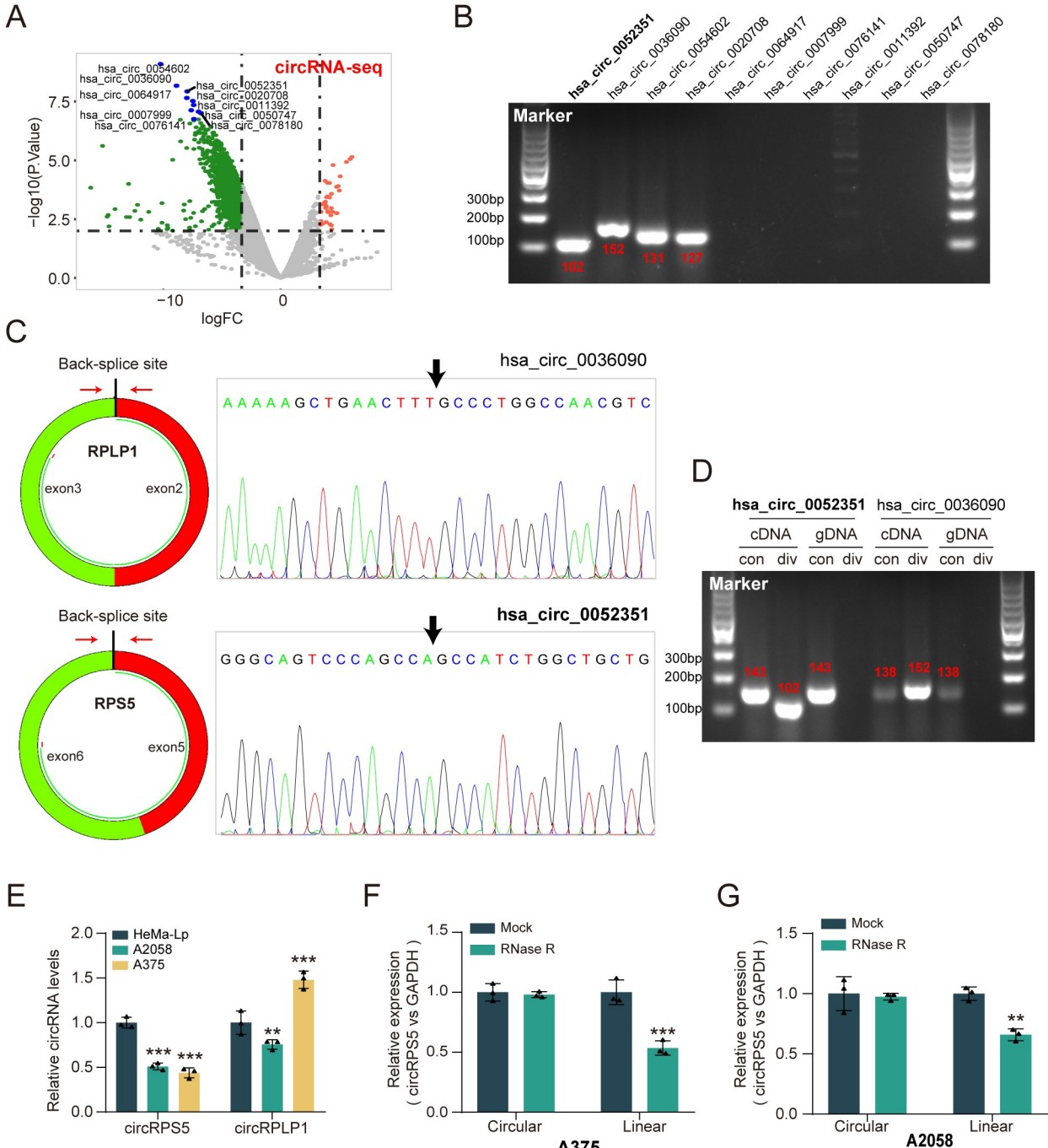

**Fig 1. Screening circRNA candidates in melanoma and characteristics of circRPS5. (A)** Volcano plot showed the differentially expressed circRNAs (DECs) in melanoma tissues using bioinformatics analysis based on circRNA-seq data. **(B)** Results of DNA electrophoresis with the PCR products of the top 10 DECs in melanoma cell lines in A375 cells. **(C)** Schematic diagram illustrated the formation of hsa_circ_0036090 and hsa_circ_0052351 originated from RPLP1 and RPS5 pre-mRNA, respectively. Sanger sequencing illustrated the joint site of circRPLP1 and circRPS5. **(D)** RT-PCR products using divergent primers indicating circularization of hsa_circ_0036090 and hsa_circ_0052351. cDNA represents complementary DNA. gDNA represents genomic DNA. **(E)** qRT-PCR revealed the expression of circRPLP1 and circRPS5 in the melanoma cell lines. **(F & G)** qRT-PCR showed the expression of circRPS5 and RPS5 mRNA in A375 cells and A2058 cells administered with RNase R or Mock control.

A2058 cells, the linear form of RPS5 was significantly reduced under RNase R treatment compared with mock, while circRPS5 was significantly resistant to RNase R (Fig 1F and 1G). Taken together, these findings indicated that circRPS5 was downregulated in melanoma tissue and cells, and was a stable and circular transcript.

## CircRPS5 functions as a tumor suppressive molecule in melanoma

In order to investigate the biological functions of circRPS5 in melanoma, we first constructed the circRPS5 overexpression system in A375 cell line by using Lv-circRNA and the circRPS5 knockdown system by using sh-circRNA in A2058 cell line. The efficiency of transfection was confirmed by qRT-PCR (Fig 2A). The CCK-8 assay showed that circRPS5 overexpression inhibited the viability of melanoma cells, however, circRPS5 knockdown significantly enhanced cell viability (Fig 2B). Consistently, the colony formation assay showed that circRPS5 overexpression significantly decreased colony forming numbers, and circRPS5 knockdown promoted it (Fig 2C). Moreover, the EdU incorporation assay also demonstrated that overexpression of circRPS5 could attenuate proliferation of melanoma cells, and circRPS5 knockdown showed the opposite result (Fig 2D). The transwell invasion and migration assay demonstrated that circRPS5 overexpression significantly reduced migration ability and invasion capacity of melanoma cells, while circRPS5 knockdown enhanced it (Fig 2E). In addition, the results of wound healing assay confirmed the effect of circRPS5 on the migration ability of melanoma cells (Fig 2F). In addition, the biomakers of cell invasion (E-cadherin and N-cadherin) were regulated by circRPS5. As shown in Fig 3C, circRPS5 overexpression significantly increased E-cadherin expression and inhibited N-cadherin expression in melanoma cells. However, N-cadherin expression was promoted and E-cadherin expression was attenuated after circRPS5 knockdown. Therefore, these findings suggest that circRPS5 could suppress migration and invasion of melanoma cells.

Subsequently, the effects of circRPS5 overexpression or knockdown on melanoma cell cycle and apoptosis were detected by flow cytometry and TUNEL staining, respectively. The results showed that circRPS5 overexpression reduced the proportion of S-phase cells in melanoma cells. Comparatively, circRPS5 knockdown produced the reverse effect (Fig 3A). In TUNEL staining, the percentage of TUNEL positive cells was increased by overexpression of circRPS5, while it was reduced by knockdown of circRPS5 (Fig 3B), which was consistent with the effect of circRPS5 on caspase3/9 expression (Fig 3C). Overall, these results demonstrated that circRPS5 halted cell cycle in G1/S transition, and promoted apoptosis of melanoma cells.

## CircRPS5 directly targets miR-151a in melanoma cells

To further understand the underlying molecular mechanism of circRPS5, we examined the subcellular localization of circRPS5 by cellular RNA fractionation and FISH assays. We found that circRPS5 was primarily distributed in the cytoplasm of melanoma cells (Fig 4A and 4B). We then used RIP assay to verify the possibility of circRPS5 binding to miRNA in melanoma cells. As shown in Fig 4C, the enrichment of circRPS5 in Ago2 group was significantly higher than that in IgG group. In order to predict the possible miRNA target of circRPS5, we overlapped the potential target miRNAs based on StarBase database with OS-related miRNAs. As a result, two potential miRNAs, hsa-miR-151a and hsa-miR-324, were identified (Fig 4D). Next, we examined the expression levels of these two miRNAs after circRPS5 overexpression and knockdown in melanoma cells. The qRT-PCR results showed the circRPS5 overexpression reduced the expression of miR-151a, and circRPS5 knockdown increased the expression of

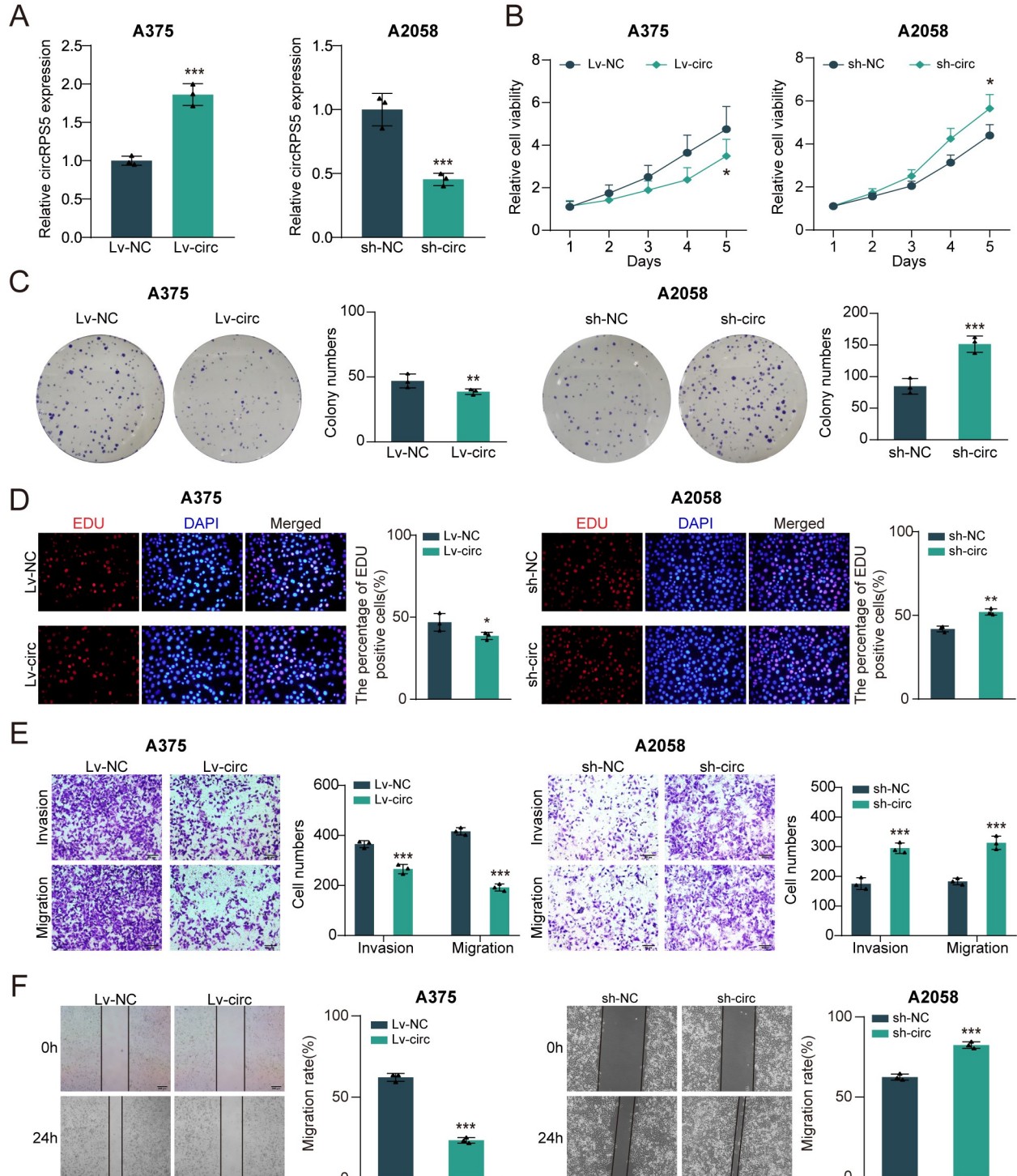

**Fig 2. CircRPS5 was a candidate melanoma suppressor in vitro. (A)** Stable oligonucleotides transfections were constructed, including circRPS5 overexpression (lv-circRPS5) in A375 cells and circRPS5 knockdown (sh-circRPS5) in A2058 cells. **(B-D)** The melanoma cell proliferation with the transfection of lv-circRPS5 and sh-circRPS5 was determined by CCK-8 (B), colony formation (C), and EdU incorporation assay (D). **(E, F)** The melanoma cell invasion and migration with the transfection of overexpression and circRPS5 knockdown was determined by transwell assay (E) and wound healing assay (F).

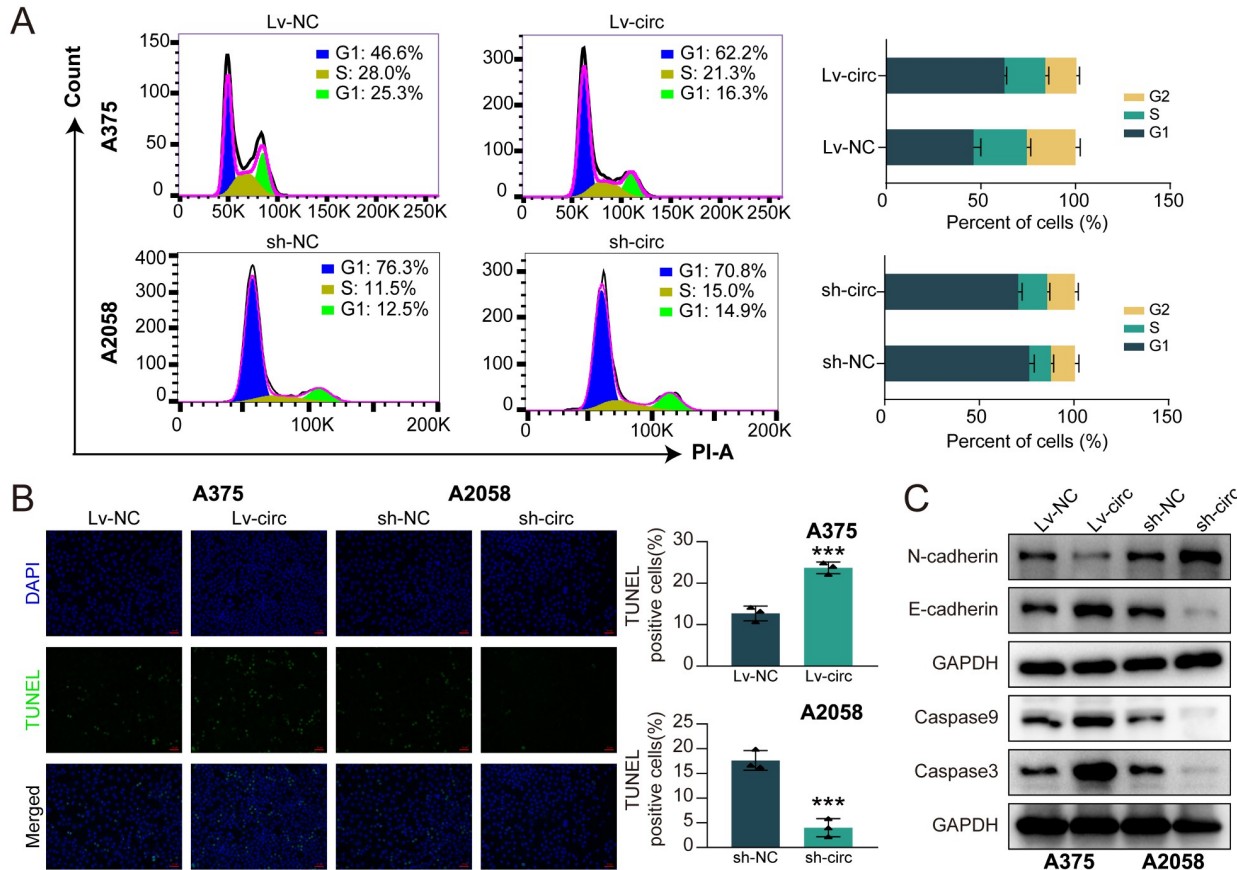

**Fig 3. CircRPS5 overexpression blocks cell cycle and induces cell apoptosis in melanoma.** (A) Cell cycle was measured by flow cytometry analyses with circRPS5 overexpression or circRPS5 knockdown in A375 and A2058 cells. (B) The TUNEL staining assay indicated the apoptotic rate of A375 and A2058 cells. (C) The expression of E-cadherin, N-cadherin, caspase3 and caspase9 was tested by western blot.

miR-151a (Fig 4E). While, there was no change in expression of miR-324 (Fig 4F). Thus, these data suggested that miR-151a might be the target of circRPS5, and the potential binding site between circRPS5 and miR-151a was shown in Fig 4G. In addition, the colocalization of circRPS5 and miR-151a in the cytoplasm of melanoma cells (Fig 4H). Of note, in dual-luciferase reporter assay, the relative luciferase activity of circRPS5 wild type (WT) and miR-151a mimic group was significantly lower than that of other groups (Fig 4I and 4J). The higher level of miR-151a expression had a lower survival rate in melanoma patients (Fig 4K). We subsequently investigated whether circRPS5 plays a tumor suppressor role in melanoma by sponging miR-151a, A375 and A2058 cells were co-transfected with circRPS5 overexpression vectors and miR-151a mimics (Fig 4L). Importantly, miR-151a upregulation blocked the inhibition of cell proliferation, migration and invasion ability caused by circRPS5 overexpression in melanoma cells (Fig 4M–4P). Together, these results suggested that circRPS5 can function as a sponge for miR-151a in melanoma.

## NPTX1 is a direct target of miR-151a

To better understand the underlying mechanisms of miR-151a, the bioinformatics databases (StarBase, GSE31909, and GSE35388) were used to predict potential target genes of miR-151a.

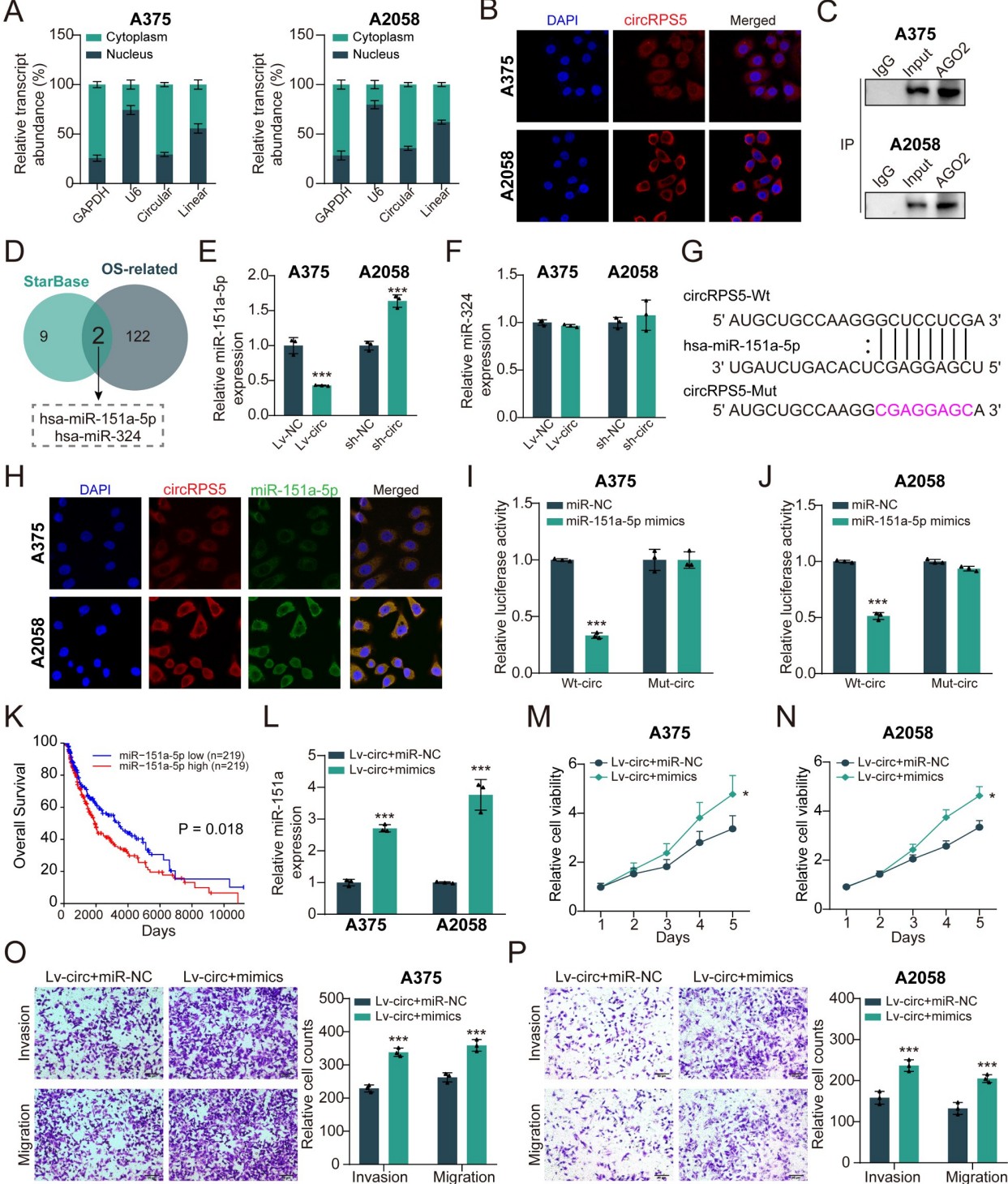

**Fig 4. CircRPS5 acts as the sponge of miR-151a. (A)** The subcellular location analysis indicated the distribution of circRPS5. **(B)** The localization of circRPS5 in melanoma cells was detected by RNA FISH. **(C)** RIP assay was used to test the combination of circRPS5 and miRNAs. **(D)** Overlap region showed the predicted miRNAs from Starbase and OS-related. **(E)** qRT-PCR analysis of miR-151a expression levels with circRPS5 overexpression and circRPS5 knockdown in A375 and A2058 cells. **(F)** qRT-PCR analysis of miR-324 expression levels with circRPS5 overexpression and circRPS5 knockdown in A375 and A2058 cells. **(G)** The potential binding site of circRPS5 and miR-151a. **(H)** RNA FISH assay was used to detect the localization of circRPS5 and miR-151a in melanoma cells. **(I, J)** Dual-luciferase reporter assay was performed to confirm the binding site of circRPS5 and miR-151a in melanoma cells. **(K)** Melanoma patients with high miR-151a expression had lower overall survival rates than patients with low miR-151a expression did. **(L)** qRT-PCR analysis of miR-151a expression in A375 and A2058 cell lines with circRPS5

overexpression or circRPS5 overexpression+ miR-151a mimics. **(M-P)** A375 and A2058 cells transfected with lv-circRPS5+miR-control, or lv-circRPS5 + miR-151a. Then the ability of cell proliferation, invasion and migration was, respectively, assessed by CCK-8 assay (M, N), transwell matrigel invasion and migration assay (O, P).

As demonstrated in the Venn diagram, a total number of 8 genes were identified as the underlying target of miR-151a (Fig 5A). Among them, 6 genes (NPTX1, AUTS2, RPS6KA2, KAT2B, MYO5A and HMG20B) were simultaneously downregulated in the melanoma samples compared with the noncancerous samples in GSE31909 and GSE35388 microarray data (Fig 5B and 5C). Then, the expression levels of these genes in melanoma cells were analysis by qRT-PCR assay. Compared with the HeMa-Lp cells, only NPTX1 expression was down-regulated in the both melanoma cells (Fig 5D). Moreover, the expression of NPTX1 was detected after circRPS5 overexpression and knockdown in melanoma cells, respectively. The results revealed that the circRPS5 overexpression enhance the NPTX1 mRNA and protein level, and circRPS5 knockdown could reduce the NPTX1 mRNA and protein level (Fig 5E and 5F). Besides, the NPTX1 mRNA has potential binding sites for miR-151a (Fig 5G). Dual-luciferase reporter assay showed that the relative luciferase activity of NPTX1 mRNA WT and miR-151a mimic groups was significantly lower than that of other groups (Fig 5H). qRT-PCR and western blotting results indicated that miR-151a mimics significantly decreased NPTX1 expression, whereas miR-151a inhibitors markedly enhanced NPTX1 expression in melanoma cells (Fig 5I and 5J). Rescue experiments revealed that miR-151a inhibitor partially reversed the inhibitory effect of circRPS5 knockdown on NPTX1 and E-cadherin expression, and reversed the promotion effect of circRPS5 knockdown on N-cadherin expression (Fig 5K). Taken together, these results suggested that NPTX1 was a target of miR-151a, and the expression of NPTX1 was also regulated by circRPS5.

## CircRPS5 inhibits cell proliferation and migration of melanoma via upregulation of NPTX1

To further confirm whether circRPS5 plays an inhibitory role in OS by up-regulating NPTX1 expression, we transfected NC, pc-NPTX1, sh- NPTX1, pc-NPTX1 + sh-circRPS5, and sh-NPTX1 + lv-circRPS5 into melanoma cells. Importantly, the expression of NPTX1 is negatively regulated by circRPS5 in melanoma cells (Fig 6A and 6B). CCK-8 and colony formation assays in melanoma cells showed that overexpression of NPTX1 led to inhibition of the proliferation capabilities, but this effect could be partly attenuated by circRPS5 knockdown. In contrast, knockdown of NPTX1 promoted cell proliferation capacity, and this effect was reversed by circRPS5 overexpression (Fig 6C and 6D). Similarly, EdU positive cells were decreased by overexpression of NPTX1, while once circRPS5 was knocked down, above impacts could be hindered. Conversely, the increased EdU positive cells after NPTX1 knockdown in melanoma cells were largely reversed by circRPS5 overexpression (Fig 6E). Furthermore, transwell assay revealed that NPTX1 overexpression obviously impeded melanoma cells invasion and migration, and this effect could be partly attenuated by circRPS5 knockdown. The invasion and migration of melanoma cells were also promoted by NPTX1 knockdown; however, circRPS5 overexpression hindered these influences (Fig 6F). Additionally, NPTX1 was found to induce cell apoptosis in melanoma cells, which was restored after circRPS5 knockdown. Whereas, knockdown of NPTX1 inhibited cell apoptosis in melanoma cells, these effects were largely abolished by circRPS5 overexpression (Fig 6G). Altogether, these data suggested that circRPS5 suppressed the progression of melanoma cells by upregulating the expression of NPTX1.

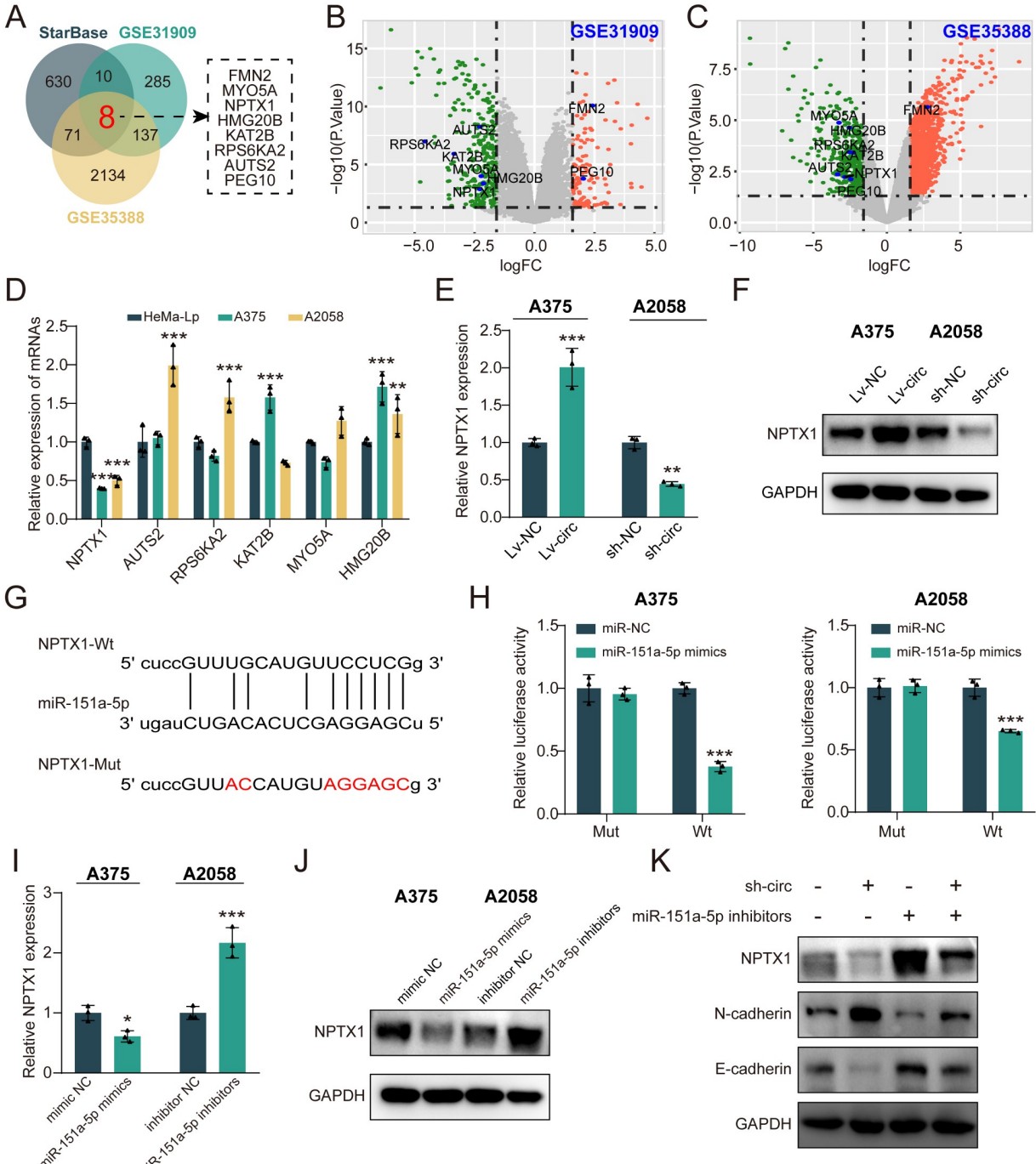

**Fig 5. NPTX1 is a direct target of miR-151a in melanoma.** (**A**) Overlapping analysis of differentially expressed genes (DEGs) in melanoma tissues using bioinformatics analysis based on StarBase, GSE31909 and GSE35388 microarray database. (**B, C**) Volcano plot of the selected 8 DEGs in the GSE31909 and GSE35388 dataset. (**D**) qRT-PCR analysis for 6 screened mRNAs in A375 and A2058 and HeMa-Lp cells. (**E**) qRT-PCR analysis showed the NPTX1 mRNA level with the transfection of circRPS5 overexpression and circRPS5 knockdown in A375 and A2058 cells. (**F**) Western blot analysis showed the NPTX1 expression level with the transfection of circRPS5 overexpression and circRPS5 knockdown in A375 and A2058 cells. (**G**) Schematic diagram indicated the targeting of miR-151a towards 3'-UTR of NPTX1 mRNA. (**H**) Dual-luciferase reporter assay was performed to confirm the binding site of miR-151a and NPTX1 mRNA in melanoma cells. (**I**) qRT-PCR analysis showed the NPTX1 mRNA level with the transfection of miR-151a mimics and miR-151a inhibitors in A375 and A2058 cells. (**J**) Western blot analysis showed the NPTX1 expression level with the transfection of miR-151a mimics and miR-151a inhibitors in A375 and A2058 cells. (**K**) Western blot analysis showed the NPTX1, E-cadherin and N-cadherin protein with transfection of sh-circRPS5 and miR-151a inhibitors.

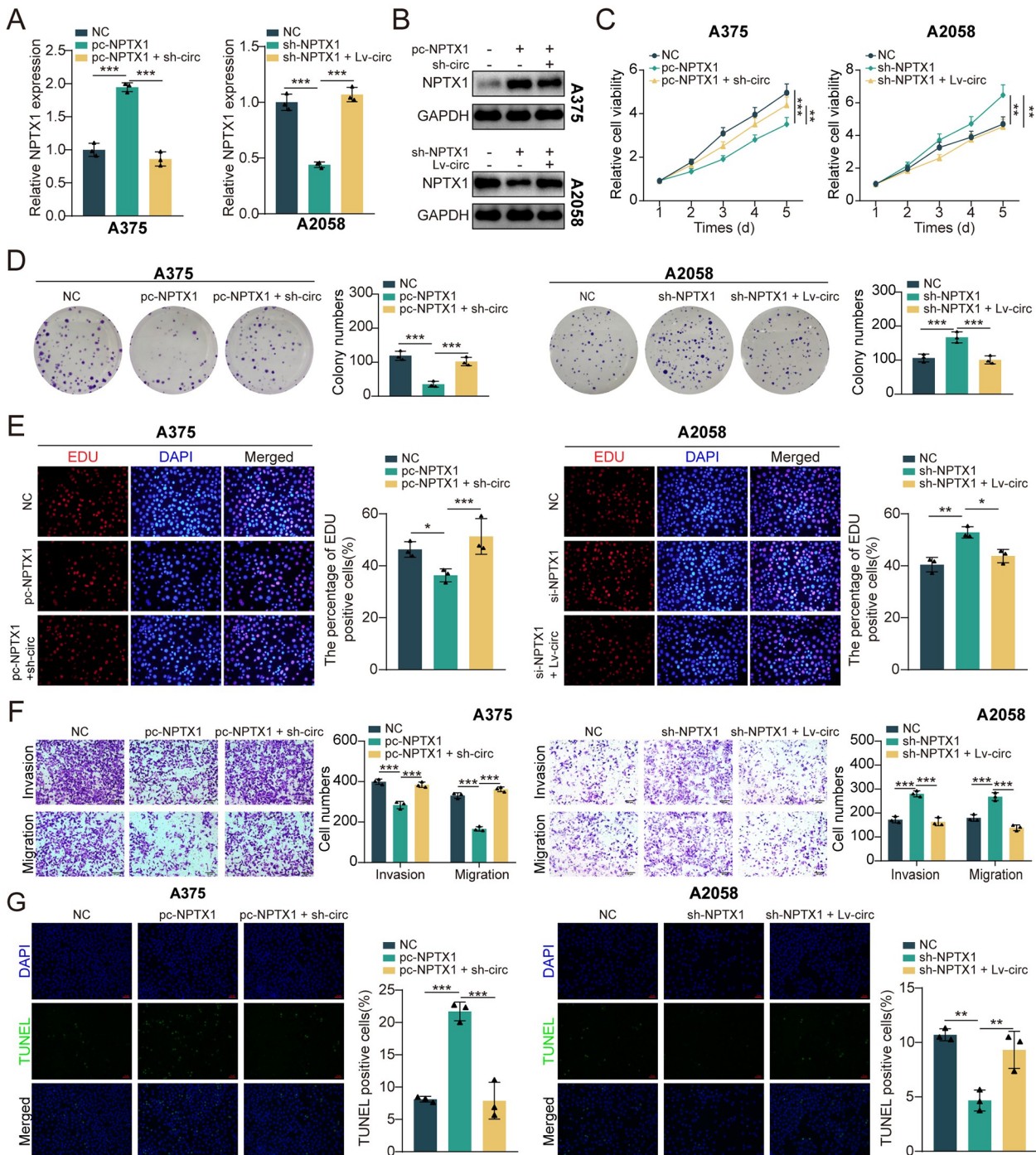

**Fig 6. CircRPS5 inhibits cell proliferation and migration of melanoma via upregulation of NPTX1. (A, B)** The melanoma cells were co-transfected with pc-NPTX1 + sh-circRPS5 or sh-NPTX1 + lv-circRPS5, the expression of NPTX1 was determined using qRT-PCR (A) and western blotting (B). **(C-E)** Cell proliferation abilities were determined by CCK-8 assay (C), colony formation assay (D), and EdU incorporation assay (E). **(F)** Cell migration and invasion abilities were checked by transwell assay. **(G)** The TUNEL staining assay indicated the apoptotic rate of melanoma cells.

## CircRPS5 reverses melanoma cell progression via incorporating into exosomes

It has been widely demonstrated that exosomes, as carriers of genetic information including circRNAs, which play a crucial role in the process of cancer development [22, 23]. Next, we explored whether extracellular circRPS5 regulated melanoma progression by incorporating into exosomes. The expression level of circRPS5 in the medium collected from the melanoma cells did not change in response to RNase A treatment, but it was significantly decreased in response to both RNase A and Triton-100 treatment, indicating that circRPS5 was encapsulated by the membrane rather than released directly (Fig 7A). Moreover, we further examined the size of these exosomes by the nanoparticle tracking analysis (NTA). The results show that the size of these exosomes comes from a similar distribution, mostly between 30 nm and 200 nm (Fig 7B). Also, the exosomal markers TSG101and HPS70 were positively expressed in A375-EXO and A2058-EXO, but not in the cell extracts (Fig 7C). In addition, the expression level of circRPS5 in exosomes was almost equal to that of extracellular circRPS5, indicating that extracellular circRPS5 was mainly incorporated into exosomes (Fig 7D). The circRPS5 levels in HeMa-Lp exosomes were significantly higher than those of A375 and A2058 exosomes (Fig 7E). We next examined whether melanoma cells could take up exosomes. As shown in Fig 7F, the results showed that HeMa-Lp-derived exosomes (PKH26-labeled) were all localized in the cytoplasm of melanoma cells, and the labeled exosomes gathered around the nucleus in the form of dots or clots, indicating that the exosomes secreted by HeMa-Lp had been successfully absorbed by melanoma recipient cells. Subsequently, the expression of circRPS5 in HeMa-Lp cell-derived exosomes was significantly decreased after circRPS5 knockdown (Fig 7G). After co-culturing with the EXO-circRPS5, the cell viability and/or invasion of melanoma cells were significantly decreased in a dose-dependent maner (Fig 7H–7J). Finally, we explored whether exosomal circRPS5 played a deterministic role, and then used GW4869 to block the production of exosomes in HeMa-Lp cells (Fig 7K and 7L). However, when HeMa-Lp cells were treated with GW4869, the HeMa-Lp-derived exosomes could hardly play the role of promoting proliferation and invasion (Fig 7M and 7N). Collectively, these results suggested that the extracellular circRPS5 inhibited melanoma cell progression through exosomes (Fig 8).

## Discussion

Mounting evidences indicate that circRNAs have a crucial role in the occurrence and progression of multiple cancers [24]. However, their function roles in melanoma remains largely unclear. In present study, we identified a novel circRNAs named circRPS5 that was significantly downregulated in melanoma tissue and cell lines. Functionally, overexpression of circRPS5 had a strong inhibitory effect on the proliferation, migration and invasion of melanoma cells, and knockdown of circRPS5 had a significant promotion effect. Mechanistically, circRPS5 could rescue the expression of NPTX1 by sponging miR-151a, thereby inhibiting melanoma progression. Moreover, circRPS5 was mainly packaged into exosomes and exerted the biological functions role in melanoma. Thus, our study demonstrated that circRPS5 may serve as a promising treatment target for patients with melanoma.

An increasing evidence has underscored that the role of circRNAs acting as competing endogenous RNAs (ceRNAs) in the development and progression of human cancers [25]. CircRNAs play as ceRNAs to regulate the protein-coding gene expression by competitive binding with miRNAs [26]. For instance, circCCDC9 was recently found to function as a ceRNA to inhibit tumor growth and metastasis in gastric cancer [27]. CircESRP1 served as an oncogenic circRNA promotes the proliferation and metastasis of endometrial cancer cells through the miR-874-3p/CPEB4 axis [28]. In this study, gain- and loss-of-function studies showed that

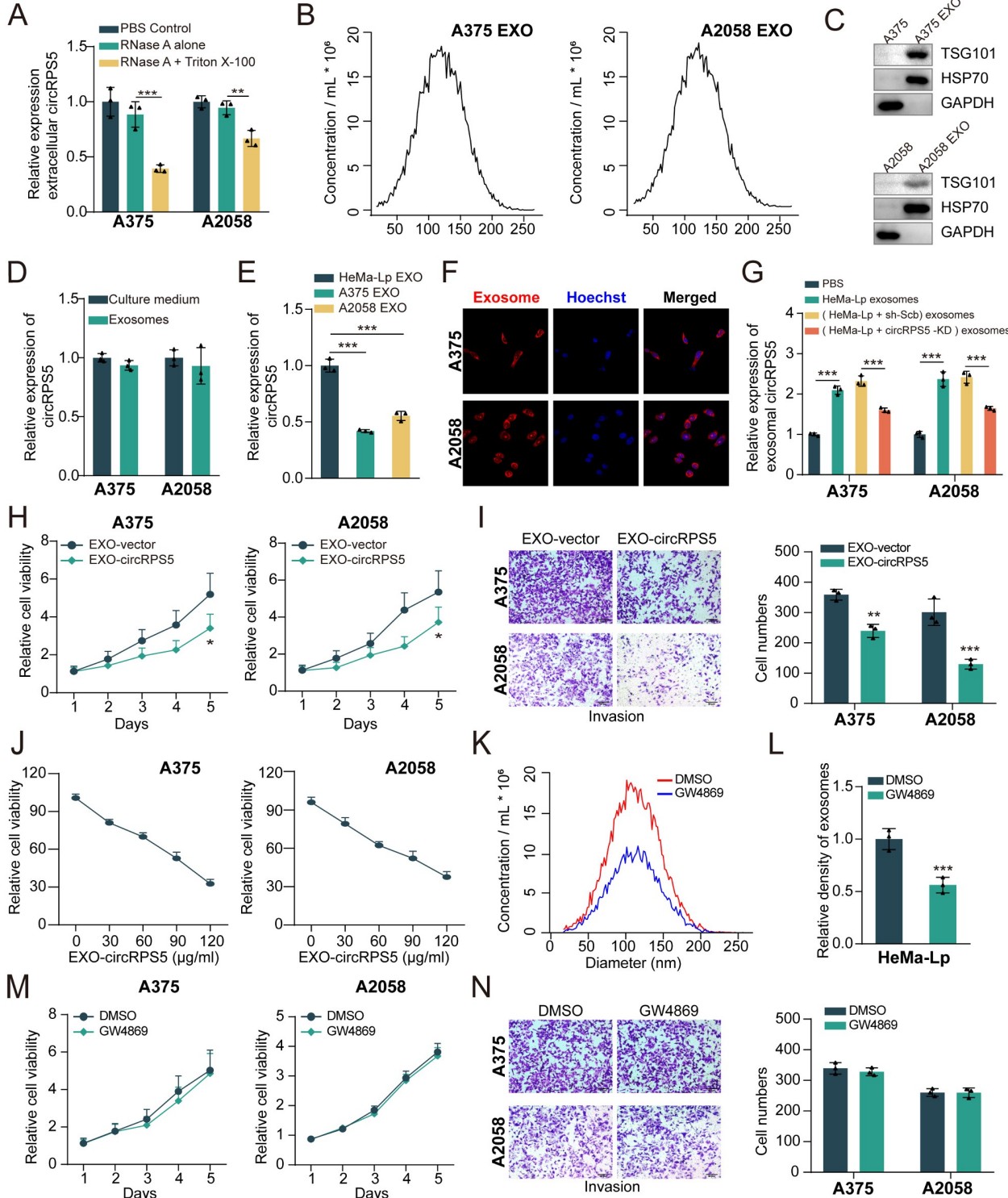

**Fig 7. CircRPS5 reverses melanoma cell progression via incorporating into exosomes. (A)** qRT-PCR showed the expression of circRPS5 in melanoma cells administered with PBS control, RNase A alone, or RNase A + Triton X-100. **(B)** Exosomes were isolated from the supernatant of the culture medium of A375 and A2058 cells, and the size range of exosomes checked by NAT analysis. **(C)** Melanoma cell-derived exosomes (A375-EXO and A2058-EXO) were analyzed by western blotting using anti-TSG101 and anti-HSP70 antibodies. Cellular lysates (A375 and A2058) were used as positive loading controls. **(D)** qRT-PCR showed the expression of circRPS5 in the supernatant of the culture medium of melanoma cells and melanoma cell-derived exosomes. **(E)** qRT-PCR showed the expression of circRPS5 in melanoma cell-derived exosomes (A375-EXO and A2058-EXO) and HeMa-Lp cell-derived exosomes (HeMa-Lp-EXO). **(F)** Confocal microscopy image of the internalization of fluorescently labelled

exosomes in A375 and A2058 cells. **(G)** The HeMa-Lp cells were transfected with sh-Scb or sh-circRPS5, and the expression of circRPS5 in HeMa-Lp cell-derived exosomes and PBS were determined using qRT-PCR. **(H)** Cell proliferation was determined by CCK8 assay. A375 and A2058 cells were treated with or without exosomes containing circRPS5 (EXO-circRPS5). **(I)** Cell invasion abilities were checked by transwell assay. **(J)** Cell viability was checked by CCK8 assay with different dose of EXO-circRPS5. **(K)** NAT analysis showed the concentration of exosomes derived from HeMa-Lp cells after treating with DMSO control or GW-4869. **(L)** The density of exosomes derived from HeMa-Lp cells administered with DMSO control or GW-4869. **(M)** Cell proliferation was determined by CCK8 assay. The A375 and A2058 cells were treated with exosomes derived from HeMa-Lp cells pretreated with DMSO control or GW-4869. **(N)** Cell invasion abilities were determined by transwell assay.

circRPS5 regulates melanoma cells proliferation, migration and invasion in vitro. Moreover, we demonstrated that circRPS5 was predominantly localized in the cytoplasm and able to bind to miR-151a. We also found that the expression levels of miR-151a were negatively correlated with the level of circRPS5. Importantly, miR-151a could reverse the effect of circRPS5 on melanoma cells progression. Therefore, our results indicated that circRPS5 exerts its tumor suppressor effect by sponging miR-151a.

Previous studies have reported that miR-151a is significantly upregulated in several types of cancers and plays a role in tumor carcinogenesis [29, 30]. For instances, miR-151a is significantly increased in non-small cell lung cancer (NSCLC) tissues and blocking miR-151a expression significantly reduced the proliferation and migration ability of NCSLC cells [29]. Guo et al. found that miR-151a-5p is highly expressed in lung cancer tissues, and miR-151a-5p inhibitor could restrain the proliferative and motility potential of lung cancer cells [30]. The results of this study showed that the expression level of miR-151a in melanoma tissues was significantly increased, and the high level of miR-151a had a poor prognosis. Moreover, miR-151a mimics could reverse the inhibitory effect of circRPS5 overexpression on the proliferation, migration and invasion of melanoma cells. Therefore, miR-151a plays a promoting role in melanoma progression. Additionally, we revealed that NPTX1 was the target of miR-151a. NPTX1 gene encodes the neuronal pentraxin 1, a member of the pentraxins family [31]. Recently, NPTX1 has been found to have decreased expression in diverse cancers [32, 33], but its expression and impact in melanoma has not been defined yet. In this study, we discovered that circRPS5 knockdown could reduce NPTX1 expression. In addition, NPTX1 knockdown significantly reversed the circRPS5 overexpression induced promotion of proliferation, invasion and migration of melanoma cells. Taken together, these data support that circRPS5 as a ceRNA to promote NPTX1-mediated proliferation and metastasis in melanoma by sponging miR-151a.

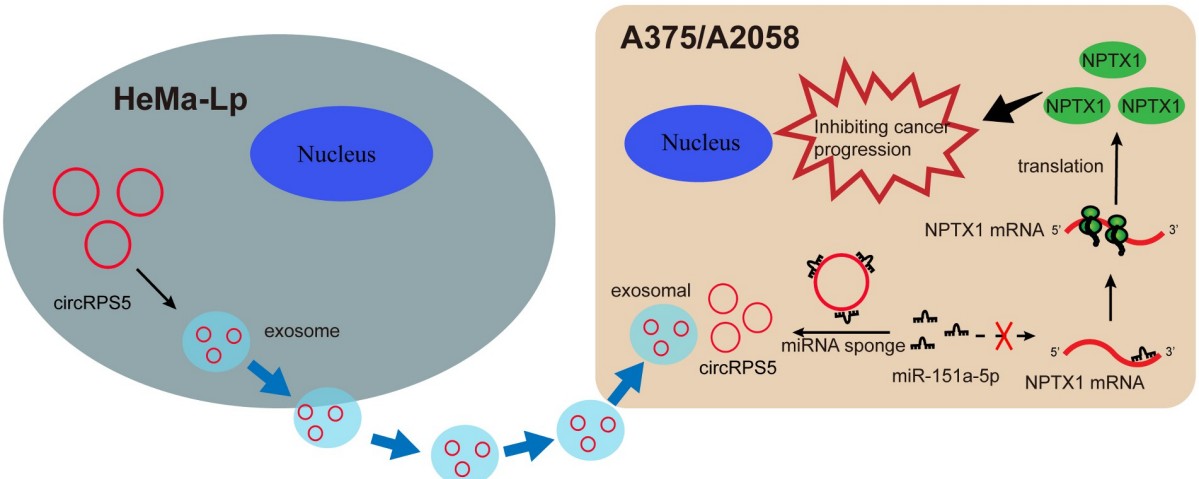

**Fig 8. The mechanisms underlying circRPS5–inhibited cancer progression.**

Exosome derived from cancer cells can modulate crosstalk with surrounding cells to remodel the tumor microenvironment, thereby enhancing cancer development and propagation [34]. Accumulating studies have reported that exosome-encapsulated circRNAs play crucial regulatory roles in the development and progression of cancers [10, 35]. For example, colorectal cancer (CRC)-derived exosomal circPACRGL promote CRC cell proliferation, migration and invasion via miR-142-3p/miR-506-3p-TGF-β1 axis [22]. In our study, we found that circRPS5 was enriched in HeMa-Lp-derived exosomes. The biological functions of circRPS5 in melanoma are mainly incorporated into exosomes.

In conclusion, the present study demonstrated that circRPS5 is significantly downregulated in melanoma tissues and cells. Overexpression of CircRPS5 can inhibit the proliferation, migration and invasion of melanoma cells via the miR-151a/NPTX1 pathway. Finally, circRPS5 could be packaged into exosomes, thereby promoting the occurrence and malignant progression of melanoma. Our findings suggest that circRPS5 could be a promising therapeutic target to prevent melanoma development and metastasis.

## Supporting information

**S1 Raw images.**
(PDF)

## Author Contributions

**Conceptualization:** Haijun Zhu, Xinping Bai.

**Data curation:** Haijun Zhu, Pan Zhang.

**Formal analysis:** Haijun Zhu, Pan Zhang.

**Funding acquisition:** Xinping Bai.

**Investigation:** Pan Zhang, Deqiang Kou, Xinping Bai.

**Methodology:** Pan Zhang, Jia Shi, Deqiang Kou.

**Project administration:** Xinping Bai.

**Resources:** Pan Zhang, Deqiang Kou, Xinping Bai.

**Software:** Jia Shi, Deqiang Kou.

**Supervision:** Jia Shi, Xinping Bai.

**Validation:** Pan Zhang, Jia Shi, Deqiang Kou.

**Visualization:** Jia Shi, Deqiang Kou.

**Writing – original draft:** Haijun Zhu.

**Writing – review & editing:** Haijun Zhu, Deqiang Kou, Xinping Bai.

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
