## [Decision Letter · Decision Letter 0]

11 Apr 2023

PONE-D-23-02043Exosome-Delivered circRPS5 Inhibits the Progression of melanoma via Regulating the miR-151a/NPTX1 AxisPLOS ONE

Dear Dr. Zhu,

Thank you for submitting your manuscript to PLOS ONE. After careful consideration, we feel that it has merit but does not fully meet PLOS ONE’s publication criteria as it currently stands. Therefore, we invite you to submit a revised version of the manuscript that addresses the points raised during the review process.

We look forward to receiving your revised manuscript.

Kind regards,

Julie Decock, PhD

Academic Editor

PLOS ONE

Journal Requirements:

Reviewers' comments:

Reviewer's Responses to Questions

**Comments to the Author**

1. Is the manuscript technically sound, and do the data support the conclusions?

Reviewer #1: Yes

Reviewer #2: Yes

2. Has the statistical analysis been performed appropriately and rigorously? 

Reviewer #1: Yes

Reviewer #2: Yes

3. Have the authors made all data underlying the findings in their manuscript fully available?

Reviewer #1: Yes

Reviewer #2: Yes

4. Is the manuscript presented in an intelligible fashion and written in standard English?

Reviewer #1: Yes

Reviewer #2: Yes

5. Review Comments to the Author

Reviewer #1: In this paper entitled “Exosome-delivered circRPS5 inhibits the progression of melanoma via regulating the miR-151a/NPTX1 axis” the authors have investigated the role of a novel exosome-delivered circular RNA named circRPS5 that plays a regulatory role in melanoma progression as well as proliferation, migration and invasion through the miR151a/NPTX1 pathway.

The study is well performed and the results are very interesting to be potentially useful for clinical application suggesting that circRPS5 could be a novel therapeutic target for melanoma therapy.

However the authors should respond to minor points. Particularly:

1) the authors through qRT-PCR demonstrate that circRPS5 is downregulated in two melanoma cell lines, A375 and A2058. In order to investigate the biological functions of circRPS5 in melanoma, the authors apply the circRPS5 overexpressing system in A375 cell lines and the circRPS5 knockdown system in A2058 cell line. Why the authors decide to proceed differently in the two cell lines of melanoma although in Fig.1E they show that the level of downregulation of the circRPS5 is equal in the two cell lines?

2) In the section Materials and Methods, in particular in the paragraph referring to the western blotting method, the authors write: "Equal amounts of protein were separated by SDS-PAGE gel...". Authors should specify the amount of protein loaded.

In conclusion the manuscript could be accepted for publication on PLOS ONE after minor revision.

Reviewer #2: The study demonstrates that circRPS5 acts as a sponge for miR-151a, which is involved in regulating the expression of the NPTX1 gene, and that circRPS5-miR-151a-NPTX1 pathway plays a crucial role in melanoma progression. The study also illustrates that circRPS5 is mainly incorporated into exosomes, which are small membrane-bound vesicles that can be secreted by cells and play a crucial role in cell-to-cell communication, and can inhibit melanoma cell progression.

Overall, the study provides new insights into the role of circRNAs in melanoma progression and identifies a potential therapeutic target for the treatment of melanoma. The findings in this study are of potential interest, but there are several issues that need to be addressed:

1. To increase the accuracy of this research, it would be beneficial for the authors to incorporate clinical samples.

2. CircRPS5 has been reported to have both tumor-suppressive and oncogenic roles in different types of cancer. Please explain the different expression of circRPS5 in tumors.

3. Is the antitumor effect of EXO-circRPS5 dose-dependent? It is suggested that the author supplement experimental proof.

4. Have the authors examined the expression of total Caspase-3/9, as they assert that circRPS5 can enhance apoptosis of melanoma cells?

5. In this study, the authors are required to ascertain whether miR-151a-3p or miR-151a-5p should be considered a direct target of circRPS5. This is because a previous study detected a downregulation in the expression level of miR-151a-3p in plasma samples of metastatic melanoma patients, indicating its role as a tumor suppressor (https://www.ncbi.nlm.nih.gov/pmc/articles/PMC5345922/#).

6. Please add a molecular pathway diagram as a graphical abstract describing circRPS5-miR-151a-NPTX1 pathway as a key regulator of melanoma.

6. PLOS authors have the option to publish the peer review history of their article (what does this mean?). If published, this will include your full peer review and any attached files.

Reviewer #1: No

Reviewer #2: No

---

## [Author Response · Author response to Decision Letter 0]

15 May 2023

Reviewer #1: 

In this paper entitled “Exosome-delivered circRPS5 inhibits the progression of melanoma via regulating the miR-151a/NPTX1 axis” the authors have investigated the role of a novel exosome-delivered circular RNA named circRPS5 that plays a regulatory role in melanoma progression as well as proliferation, migration and invasion through the miR151a/NPTX1 pathway.

The study is well performed and the results are very interesting to be potentially useful for clinical application suggesting that circRPS5 could be a novel therapeutic target for melanoma therapy.

However the authors should respond to minor points. Particularly:

1) the authors through qRT-PCR demonstrate that circRPS5 is downregulated in two melanoma cell lines, A375 and A2058. In order to investigate the biological functions of circRPS5 in melanoma, the authors apply the circRPS5 overexpressing system in A375 cell lines and the circRPS5 knockdown system in A2058 cell line. Why the authors decide to proceed differently in the two cell lines of melanoma although in Fig.1E they show that the level of downregulation of the circRPS5 is equal in the two cell lines?

Response: Thank you for raising this concern. We only want to observe the biological effects of circRPS5 from the two dimensions of overexpression and knockdown, so that the results can be more reliable. One cell line in those two cell lines must overexpress circRPS5, and another cell line knockdown circRPS5, so we select the cells with high expression for knockdown, and the cells with low expression for overexpression. Although there was no significant difference in the expression of circRPS5 between the two cells, the expression of circRPS5 was indeed higher in A2058 cells than in A375, and such experimental design was reasonable.

2) In the section Materials and Methods, in particular in the paragraph referring to the western blotting method, the authors write: "Equal amounts of protein were separated by SDS-PAGE gel...". Authors should specify the amount of protein loaded.

Response: Thanks for your thoughtful question. Based on your suggestion, we have specified the amount of protein loaded and updated the revised manuscript.

In conclusion the manuscript could be accepted for publication on PLOS ONE after minor revision.

Reviewer #2: 

The study demonstrates that circRPS5 acts as a sponge for miR-151a, which is involved in regulating the expression of the NPTX1 gene, and that circRPS5-miR-151a-NPTX1 pathway plays a crucial role in melanoma progression. The study also illustrates that circRPS5 is mainly incorporated into exosomes, which are small membrane-bound vesicles that can be secreted by cells and play a crucial role in cell-to-cell communication, and can inhibit melanoma cell progression.

Overall, the study provides new insights into the role of circRNAs in melanoma progression and identifies a potential therapeutic target for the treatment of melanoma. The findings in this study are of potential interest, but there are several issues that need to be addressed:

1. To increase the accuracy of this research, it would be beneficial for the authors to incorporate clinical samples.

Response: Thanks for your thoughtful question. We have updated the verb form of research in this paper. We previously planned to collect clinical specimens to verify our molecular expression level, but the number of osteosarcoma patients is not large, and it will take a long time to collect enough samples. We will continue to collect samples in the following work to verify the expression level and prognosis of the main molecules in this paper.

2. CircRPS5 has been reported to have both tumor-suppressive and oncogenic roles in different types of cancer. Please explain the different expression of circRPS5 in tumors.

Response: Thank you for raising this concern. There are few reports about CircRPS5. One study report that CircRPS5 is related to the polarization of M2 macrophages in brain injury process (PMID: 34896256). It is a well-known fact that different tumor cells activate different genes and activate them differently at different stages of tumorigenesis. A comprehensive and extensive explanation can be found in this article entitled “Tumor Suppressors Having Oncogenic Functions: The Double Agents” (PMID: 33396222).

3. Is the antitumor effect of EXO-circRPS5 dose-dependent? It is suggested that the author supplement experimental proof.

Response: Thanks for your thoughtful question. In response to your questions, we added experiments to explore and found that the anti-tumor effect of EXO-circRPS5 is indeed dose-dependent, the experiment results have been added in the revised manuscript and figure 7J.

4. Have the authors examined the expression of total Caspase-3/9, as they assert that circRPS5 can enhance apoptosis of melanoma cells?

Response: Thanks for your thoughtful question. We added experiments to observe the effect of circRPS5 on apoptosis and found that circRPS5 can enhance apoptosis of melanoma cells, the experiment results have been added in the revised manuscript and figure 3C.

5. In this study, the authors are required to ascertain whether miR-151a-3p or miR-151a-5p should be considered a direct target of circRPS5. This is because a previous study detected a downregulation in the expression level of miR-151a-3p in plasma samples of metastatic melanoma patients, indicating its role as a tumor suppressor (https://www.ncbi.nlm.nih.gov/pmc/articles/PMC5345922/#).

Response: Thanks for your kind suggestions. We have reconfirmed in the Starbase database that the targeted molecule of circRPS5 is miR-151a-5p, not miR-151a-3p, and we have replaced the miR-151a in miR-151a-5p in the revised manuscript.

6. Please add a molecular pathway diagram as a graphical abstract describing circRPS5-miR-151a-NPTX1 pathway as a key regulator of melanoma.

Response: Thanks for your thoughtful question. We have added a graphical abstract in Figure 8.

---

## [Decision Letter · Decision Letter 1]

5 Jun 2023

Exosome-Delivered circRPS5 Inhibits the Progression of melanoma via Regulating the miR-151a/NPTX1 Axis

PONE-D-23-02043R1

Dear Dr. Zhu,

We’re pleased to inform you that your manuscript has been judged scientifically suitable for publication and will be formally accepted for publication once it meets all outstanding technical requirements.

Kind regards,

Julie Decock, PhD

Academic Editor

PLOS ONE

---

## [Editor Report · Acceptance letter]

22 Jun 2023

PONE-D-23-02043R1 

Exosome-Delivered circRPS5 Inhibits the Progression of melanoma via Regulating the miR-151a/NPTX1 Axis 

Dear Dr. Zhu:

I'm pleased to inform you that your manuscript has been deemed suitable for publication in PLOS ONE. Congratulations! Your manuscript is now with our production department. 

Kind regards, 

on behalf of

Dr. Julie Decock 

Academic Editor

PLOS ONE